# Research of Potential Catalysts for Two-Component Silyl-Terminated Prepolymer/Epoxy Resin Adhesives

**DOI:** 10.3390/polym15102269

**Published:** 2023-05-11

**Authors:** Ritvars Berzins, Remo Merijs-Meri, Janis Zicans

**Affiliations:** Institute of Polymer Materials, Faculty of Materials Science and Applied Chemistry, Riga Technical University, 3 Paula Valdena Street, LV-1048 Riga, Latvia; remo.merijs-meri@rtu.lv (R.M.-M.); janis.zicans@rtu.lv (J.Z.)

**Keywords:** two-component adhesive, catalysts, silyl-terminated prepolymer, epoxy resin

## Abstract

The current research is devoted to the research of potential catalysts for the two-component silyl-terminated prepolymer/epoxy resin system. The catalyst system must catalyze the prepolymer of the opposite component while not curing the prepolymer in the component in which the catalyst is located. Mechanical and rheological characterization of the adhesive was performed. The results of the investigation showed that certain alternative catalyst systems, which are less toxic, may be used instead of traditional catalysts for individual systems. Two-component systems, obtained by using these catalysts systems, cure in an acceptable time scale and demonstrate relatively high tensile strength and deformation values.

## 1. Introduction

Polymer adhesives are used in many important industrial sectors, including construction, automotive, electronics, packaging, shipbuilding, medicine and others. As the standard of living and population increase, the demands on materials and their quantities are increasing, which is also the reason for constant growth in the adhesive industry sector. Adhesives market turnover exceeded USD 44.1 billion in 2021 and is estimated to grow more than 6% every year until 2027 [1]. Despite numerous commercial formulations and high turnover, the industry is relatively stagnant and provides only a limited offer of polymer-based adhesives. The system studied in the work belongs to the elastomeric adhesives, the largest members of this segment are polyurethane and silyl-terminated polymers [2,3], which are available in one-component or two-component formulations. They are widely used because of the ability to ensure a broad property envelope, high reactivity at low temperatures and adhesion to various substrates [4,5,6,7,8,9]. Adhesion and material mechanical properties are largely dependent on the polymers, plasticizers, fillers and adhesion promoters present in the system, but it is very important to choose specific and effective catalyst systems to achieve good adhesion, workability time and the best possible mechanical properties ensuring the formation of a polymer network [10,11,12]. Organic catalysts containing tin mostly are used for silyl-terminated polymer systems (SIL) because they are able to harden silyl-terminated prepolymer even at low concentrations, and they are also stable in systems, which contain water [13,14,15,16,17,18]. Epoxy resins (EPs) are cured with various substituted amines, which allow them to be activated without using a functional amine connection, as a result without using a third polymer system in the formation of material properties [16,17]. In the current research, attention is devoted to research suitable catalysts for two-component SIL/EP systems [18]. The catalyst must selectively cure a particular prepolymer without solidifying the other, in the component of which it will be incorporated to form the final adhesive material. The studied catalysts were integrated into two-component SIL/EP model systems, and their mechanical properties were determined by demonstrating the efficiency of the catalysts. Several commercially important catalysts were used during the work, and their concentrations relative to the prepolymers were chosen based on the recommendations of their manufacturers. The results showed at what concentrations the catalysts worked, as well as whether less toxic catalysts worked in the SIL and EP systems and whether they are potentially usable in the studied systems.

## 2. Materials and Methods

### 2.1. Materials

Materials used for the development of the investigated SIL/EP two-component system catalysts are summarized in Table 1, whereas Table 2 shows the recipes of the model systems.

Mixtures were mixed using the SpeedMixer DAC 150 centrifugal laboratory mixer, casted in Teflon molds and cured at standard conditions—23 ± 2 °C and 50 ± 5% RH—for 1, 7 and 28 days.

### 2.2. Testing Methods

#### 2.2.1. Tensile Test

Tensile stress–strain measurements were made using the Zwick/Roell Z010 universal testing machine (ZwickRoell GmbH & Co. KG, Ulm, Germany). The tests were made according to ISO 527 at a test speed of 100 mm/min (dumbbell specimens). Testing was performed for model systems after 1, 7 and 28 days of curing at 23 °C and 50% RH.

#### 2.2.2. Hardness Test

Hardness was tested according to ISO 7619 [19], using SCHMIDT PHPSA equipment. Testing (Schmidt control instruments, Waldkraiburg, Germany) was performed for model systems after 1, 7 and 28 days of curing at 23 °C, 50% RH.

#### 2.2.3. Rheology Tests

Viscosity of prepolymers was tested using the Bohlin CVO 100 rheometer (Malvern, Grovewood Rd, UK). The instrument was equipped with a 20 mm diameter spindle with plate–plate geometry (gap size 1000 µm). Tests were made at a constant shear rate of 5 s^−1^ and a temperature of 25 °C for 300 s.

## 3. Results

### 3.1. Rheological Properties of Silyl-Terminated Prepolymer (SAX 520)

Viscosity changes in time were studied for systems consisting of individual prepolymers (SIL and EP) and a catalyst. The mixtures were immediately placed on the surface of the rheometer’s lower plate, and a viscosity change test was performed. Then, 0.2% water was added to the silyl-terminated prepolymer mixture to initiate a hydrolysis reaction followed by polycondensation of the prepolymer, which in real systems is typically obtained from moisture in the fillers. The introduced water promotes hydrolysis of the methoxy group, followed by a polycondensation reaction to form a polymer.

SAX 520 was used as a prepolymer of the silyl-terminated component. SAX 520 is a highly reactive prepolymer because it contains three reactive methoxy groups, which can react with water and then condense to form high molecular compounds. Catalyst company TIB Chemicals, which specializes in the development of catalysts for such systems, recommended using tin-type organometallic catalysts for such prepolymers at the minimum concentrations of 0.2%. Catalysis of organosilanes using tin–based compounds begins with hydrolysis. The hydrolyzed tin compound reacts with the alkoxy group to form an organotin silanolate. The organosilanolate will react with formed silanol groups producing siloxane linkages, and the organotin hydroxide catalysis is regenerated (the process is cyclical), Figure 1 [20]. The second potential curing mechanism of silylterminated prepolymers is using amine-type catalysts. Alkoxysilane hydrolysis occurs by forming pentacoordinate intermediate via two different transition states. In the presence of a base and water during the hydrolysis reaction, a partial negative charge develops on the alkoxysilane forming the first transition state. The first transition state dissociates and generates the pentacoordinate intermediate, which subsequently breaks down to the desired silanol via the second transition state (Figure 2) [20]. In general base catalysis, any basic species accelerates the reaction by the deprotonating proton forming transition state (hydroxide anion accelerates the reaction rate by directly attacking the substrate).

The results showed that all tin-type catalysts catalyzed SAX prepolymer (Figure 3a), whereas the most effective was Tibcat 318 (dioctyl tin dicarboxylate). Organotin catalysts are traditionally used for silyl-terminated one-component systems, so it is not surprising that they catalyze it; however, they must comply with the condition that they do not harden the epoxy prepolymer in order to make a two-component system, which will be stable during storage.

DBU was tested at two catalyst concentrations of 0.5 and 4%. These concentrations were chosen based on the recommendations of several catalyst manufacturers, given that amine catalysts are typically less active than organometallic ones. Mixtures of SAX 520 with amine-type catalysts showed that only one of them catalyzes the system (Figure 3b) at the highest catalyst concentration (w = 4%). The DBU catalyst practically did not catalyze the SIL prepolymer at a concentration of 0.5%, but at higher catalyst concentrations, the reactivity increased rapidly, from which it can be concluded that the DBU catalyst needs to reach a certain concentration in order to activate the hydrolysis and subsequent condensation reactions of the SIL prepolymer, making it a significant alternative to tin-type catalysts. The rest of the studied catalyst systems (Niax C41, Dabco 33LV, DMDEE, Ancamine K54) practically did not increase the viscosity during the experiment, indicating that the catalysts do not catalyze the SAX 520 prepolymer; however, it can potentially be used for epoxy resin systems, thus potentially keeping the component with the SIL prepolymer stable. By studying organometallic catalysts of bismuth, zirconium and titanium (Figure 3c), it was revealed that only titanium chelates catalyst catalyzed the SAX 520 system; however, even at higher catalyst concentrations, the viscosity increase is not fast enough to be used as a catalyst for the system. If the catalyst is inefficient, the final adhesive material will not be active enough to meet exploitation requirements, and its high catalyst concentration will make the material too expensive, especially in the case of organometallic catalysts, which are typically at least 4 times more expensive than the prepolymer.

### 3.2. Rheological Properties of Epoxy Resin (D.E.R. 331)

Primary and secondary amines are traditionally used as epoxy resin hardeners at room temperature; however, two-component SIL/EP systems have been designed to connect epoxy systems in the uniform polymer network with an SIL prepolymer, so it is necessary to catalyze epoxy resins using non-functional amine catalysts (tertiary amines). Amine attacks the oxirane ring, destabilizing it and causing it to open and form active functional groups (ions) that can react with the available groups in the systems, in this case with another epoxy compound (Figure 4) [21].

Systems with organotin catalysts do not significantly increase the viscosity during the experiment, indicating that the catalysts are inefficient for epoxy resin (D.E.R. 331) systems (Figure 5a). From the obtained results, we can conclude that tin-type catalysts can be used to produce a two-component SIL/EP system as the SIL catalyst, when incorporated into the epoxy component, does not react with the EP prepolymer.

In the case of a full two-component system, both the tertiary amine and the secondary amino silane are used to cure the epoxy prepolymer, thus using two curing mechanisms to form a uniform system with the silyl-terminated prepolymer. The results obtained (Figure 5b) show that two of the amine catalysts can be used to cure epoxy resin prepolymer (Niax C41 and Ancamine K54), and the respective catalysts have exponential curing dynamics to ensure the relatively fast formation of the polymer network. More effective was the Ancamine K54 catalyst; however, this catalyst is also more toxic compared to Niax C41. By halving the catalyst concentration, the less effective Niax C41 catalyst reduces reactivity more than Ancamine K54 (Figure 5c); however, both catalysts’ reactivity is still sufficient to cure epoxy resin systems. Both catalysts can be used to produce two-component SIL/EP systems as the D.E.R. 331 catalyst by incorporating it into the SIL component, which does not react with it. The results show that DBU does not catalyze the epoxy system but catalyzes the SIL component, making it an alternative catalyst for two-component SIL/EP systems.

Systems with organometallic catalysts (bismuth, zirconium and titanium) do not significantly increase the viscosity during the experiment (Figure 2d), indicating that the catalysts are inefficient for epoxy resin (D.E.R. 331) systems.

In the following work (model systems), we chose three types of catalysts, which, according to previous studies, can be integrated into a two-component system. These catalyst systems were Tibcat 318/Ancamine K54, Tibcat 318/Niax C41 and the less environmentally hazardous version DBU/Niax C41. Their concentrations in the composition of the investigated model systems are given in Table 2.

### 3.3. Mechanical Properties of the Two-Component Model Systems

The tensile properties of the materials (Figure 6 and Figure 7) were tested based on the results shown above for the performance of catalysts in individual prepolymer systems. Two types of catalysts, Niax C41 and Ancamine K54, were used to cure epoxy systems. The Tibcat 318 and DBU catalysts were used for the catalysis of silyl-terminated systems. Amine-type catalysts were used to cure epoxy resin (D.E.R. 331) and silyl-terminated prepolymer (SAX 520) at its maximum researched concentration (4%) by weight of the prepolymer, and tin catalysts were used to cure silyl-terminated (SAX 520) prepolymer at the lowest concentration (0.2%) by weight of the prepolymer.

All the investigated two-component SIL/EP systems with different catalysts (Ancamine K54/Tibcat 318, Niax C41/Tibcat 318, Niax C41/DBU) showed similar hardening dynamics in all the studied prepolymer ratios, showing tensile strength maximum peaks at SIL/EP ratios from 60/40 to 40/60.

Tensile strength values increase in time with an increasing degree of polymer crosslinking. After 1 day of curing at certain SIL/EP ratios, all the systems’ maximum tensile strength values exceeded 2 MPa. The highest value was the system with Niax C41/DBU catalysts, which can be explained by the fact that DBU at the concentration of 4% by weight of the prepolymer showed faster curing dynamics compared to the tin-type catalyst at lower concentrations. Considering that DBU catalyzes the SAX 520 prepolymer faster, its peak values are shifted to an SIL/EP ratio of 40/60, increasing the contribution of the SIL prepolymer in the development of the final 3D polymer network. By increasing the EP prepolymer concentration to over 60%, even the tensile strength values of the system with the Niax C41/DBU catalyst pair decreased because of the increased concentration of rigid structural elements (D.E.R. 331) in the system, which reduces its flexibility (the material turns brittle). After 7 days of curing, the highest value of tensile strength (4.53 MPa) was the system with the Ancamine K54/Tibcat 318 catalyst pair at an SIL/EP ratio of 50/50. The systems with the Niax C41/Tibcat 318 and Niax C41/DBU catalyst pairs at their maximum tensile strength showed 12% and 9% lower values, respectively. This indicates that the effect of the studied catalysts on the tensile strength properties of the material is relatively small after 7 days of curing, i.e., the mechanical properties of the systems depend more on the polymer composition than on the catalyst pair. After 28 days of curing, all the tensile strength values of the adhesive compositions increased by over 5 MPa at their maximum, indicating that all the catalytic systems are efficient and potentially will form adhesives with high tensile strength. The system with the Ancamine K54/Tibcat 318 catalyst pair showed the highest tensile strength value (6.18 MPa) after 28 days of curing at 23 °C and 50% RH. Concomitantly, the other investigated systems at their maximums showed 14% and 10% lower tensile strength values, i.e., 5.28 MPa and 5.53 MPa, respectively, for the systems with the Niax C41/Tibcat 318 and Niax C41/DBU catalyst pairs. The system with Ancamine K54/Tibcat 318 showed the highest tensile strength value because Ancamine K54 is the most effective catalyst for epoxy resin, ensuring maximum hardening for epoxide elements in the polymer network, which is explained by the structure of its aromatic ring at the substituted amine. Epoxy compounds are less reactive due to their rigid structure, and to harden completely, it is necessary to choose the most efficient catalyst system, or if that is not enough, the only option is to supply additional energy, which is usually achieved by performing curing at elevated temperatures.

The tensile deformation values of the investigated systems decreased in time with an increasing degree of polymer crosslinking. After 1 day of curing, the lowest deformation was for the Niax C41/DBU catalyzed system because the catalyst concentration of the respective system was the highest (DBU (4%) catalyst for the SAX 520 prepolymer) most effectively causing the crosslinking of the system, resulting in reduced deformation values already after 1 day of curing. The higher concentration of the catalysts also potentially promotes the formation of two separate segments of polymer networks, reducing the deformation values over the entire test range (1–28 days) compared to the other systems tested. Typically, as the systems have faster curing rates, a more chaotic polymer network forms, as it does not have time to arrange itself for potentially higher mechanical properties. During the hardening of the systems, the smallest changes were shown for the system with the Ancamine K54/Tibcat 318 catalyst pair (values difference between 1 and 28 days: Δε_Ancamine K54/Tibcat 318_ = −93%; Δε_Niax C41/Tibcat 318_ = −190%; Δε_Niax C41/DBU_ = −111%). This indicates that using this particular catalyst system forms the most flexible and uniform polymer network, which is also confirmed by respective changes in tensile strength values during the hardening period.

Figure 8 shows material hardness changes over time from 1 to 28 days. The systems with the Niax C41/Tibcat 318 and Niax C41/DBU catalyst systems showed the highest hardness values after curing for 1 day. As both systems contained the Niax C41 catalyst, it can be concluded that this catalyst promotes a larger increase in material hardness after 1 day of curing compared to Ancamine K54. Despite the fact that the concentration of the DBU catalyst is higher in comparison to Tibcat 318 (4% vs. 0.2%), the hardness of the system, catalyzed by DBU/Niax C41, after 1 day of curing practically did not differ from that of the Niax C41/Tibcat 318 system. This allows concluding that material hardness depends largely on the concentration of the rigid element (D.E.R. 331) within the system and its ability to crosslink. The slower curing rate of the epoxy compound explains why after 1 day of curing at 23 °C and 50% RH, practically the same hardness values of the investigated systems are observed, i.e., after 1 day of curing, the mechanical properties are determined practically by the crosslinking of the SIL prepolymer. All the materials increased the hardness values by increasing the curing time, which can be explained by the fact that the degree of crosslinks within the polymer network increases over time, and an increasing number of epoxide molecules integrate into the polymer network. As expected, the highest hardness values after 28 days of curing were shown for the systems obtained by using the Niax C41/DBU catalyst pair. Higher concentrations of these catalysts lead to a higher degree of crosslinking, resulting in a harder material; however, the hardness values of the systems obtained by using other catalyst pairs were only slightly lower.

### 3.4. Rheological Properties of the Model Systems

The rheological properties of the model systems at two different SIL/EP concentrations were determined using the oscillation mode. SIL/EP ratios of 60/40 and 50/50 were chosen because the maximum tensile mechanical properties were observed at these prepolymer ratios.

As the content of the EP prepolymer increases, the crossover modulus of the system increases, indicating that the system becomes stiffer, which is also indicated by the mechanical properties in tension, as well as the hardness of the materials. The system with the Tibcat 318/Ancamine K54 catalyst system showed the highest modulus value, indicating that the stiffness of the material is affected mostly by the epoxy component, resulting in an increase in material stiffness (Figure 9 and Table 3). By increasing the concentration of the EP prepolymer in the system, the crossover modulus is reached after a longer time, indicating that the curing rate of the SIL prepolymer is higher compared to the EP prepolymer. The relevant effect may be explained by the higher functionality of the SIL prepolymer. For the Tibcat 318/Niax C41 catalyst system, the time to reach the value of crossover modulus differs the least when the prepolymer ratio changes, indicating that the respective catalyst system’s curing rates are the closest, resulting in a more uniform polymer network.

## 4. Conclusions

The results of the rheological properties of different catalysts systems showed that the investigated SIL prepolymer may be efficiently catalyzed either by traditional tin-organic compounds at low concentration (0.2%) or by less harmful amine compounds (DBU) at higher catalyst concentration (4%). The EP prepolymer may be catalyzed by using amine-type catalysts, such as Niax C41 and Ancamine K54, from which the latter showed higher activity.

Integration of the most rheologically effective catalyst pairs (Ancamine K54/Tibcat 318, Niax C41/Tibcat 318 and Niax C41/DBU) into the SIL/EP model systems showed that the highest tensile strength and deformation properties after 28 days of hardening at 23 °C and 50% RH are achieved in the case of Ancamine K54/Tibcat 318 catalysts. The system with the highest catalyst concentrations (Niax C41/DBU) showed the highest material hardness values, which may be explained by the fact that the highest crosslink density is developed in the system. In spite of the higher costs of the Niax C41/DBU system, it is regarded as a better alternative from a chemical hazard standpoint.

The current research reveals the efficiency of different catalyst pairs for the development of two-component model systems based on silyl-terminated and epoxy prepolymers, which will allow elaborating commercially viable high-performance materials for the adhesives and sealants market. This work shows that it is possible to create two-component systems using only amine-type catalysts, the application of which reduces the harmfulness of the overall system, thus increasing its environmental sustainability.

## Figures and Tables

**Figure 1 polymers-15-02269-f001:**
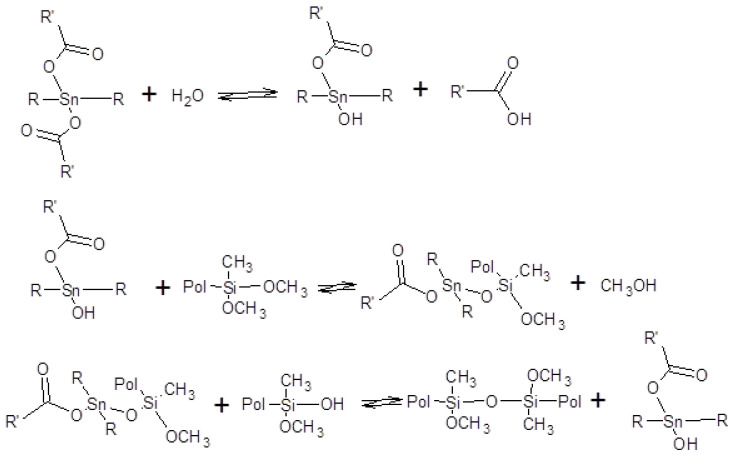
Curing mechanism of silyl-terminated prepolymers using tin-type catalysts.

**Figure 2 polymers-15-02269-f002:**
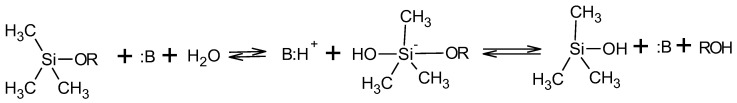
Curing mechanism of silyl-terminated prepolymers using amine-type catalysts.

**Figure 3 polymers-15-02269-f003:**
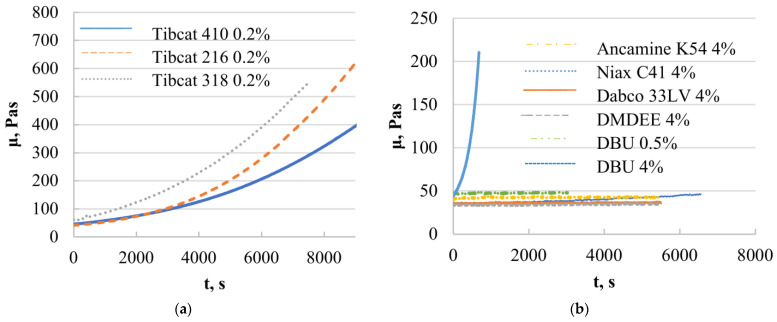
Viscosity change in time of the systems of SAX 520 prepolymer and different tin-organic (**a**), amine-containing (**b**) and organometallic (**c**) catalysts.

**Figure 4 polymers-15-02269-f004:**
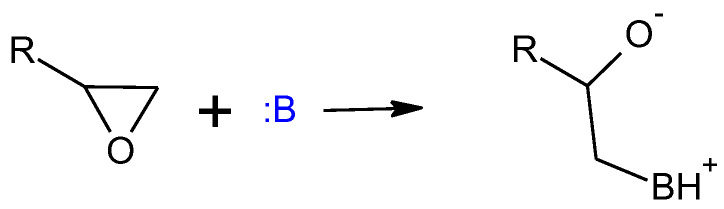
Curing mechanism of epoxy prepolymers using amine-type catalysts.

**Figure 5 polymers-15-02269-f005:**
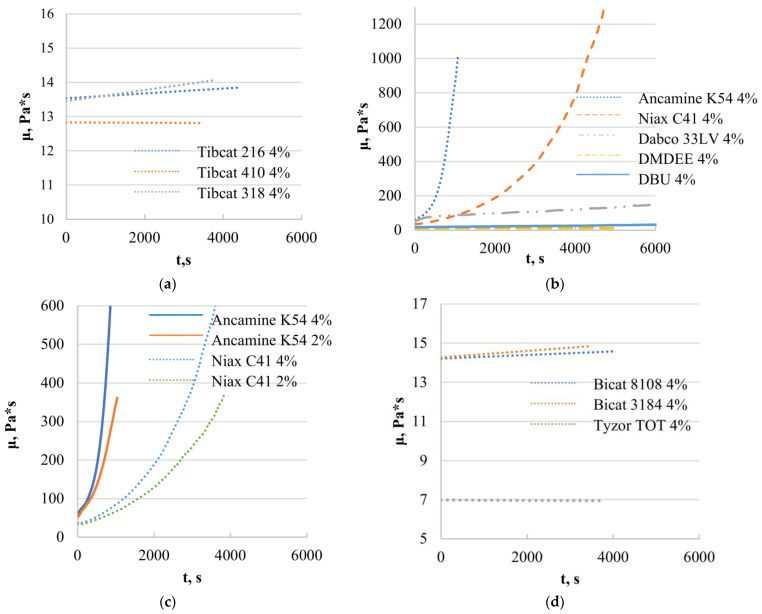
Viscosity change in time of the systems of D.E.R. 331 prepolymer and different organotin (**a**), amine-containing (**b**,**c**) and organometallic (**d**) catalysts.

**Figure 6 polymers-15-02269-f006:**
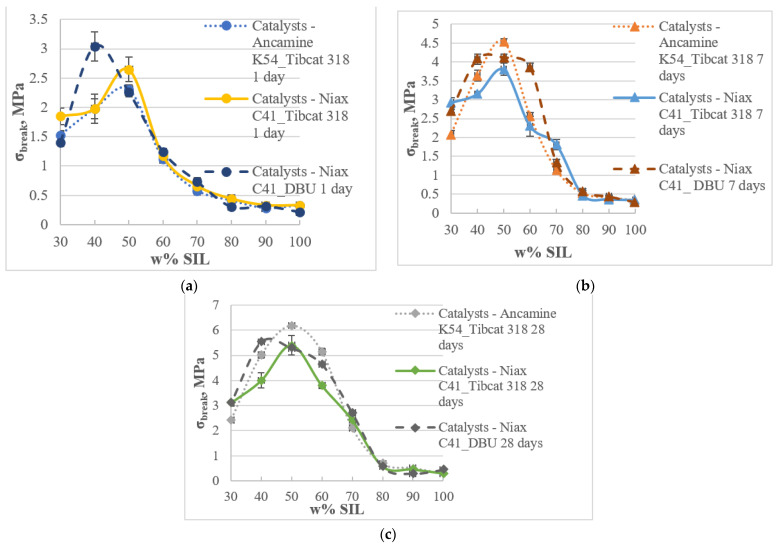
Tensile strength (σ_break_) of the SIL/EP blends with different catalysts systems: Ancamine K54/Tibcat 318, Niax C41/Tibcat 318 and Niax C41/DBU after 1 (**a**), 7 (**b**) and 28 (**c**) days of curing at 23 °C and 50% RH.

**Figure 7 polymers-15-02269-f007:**
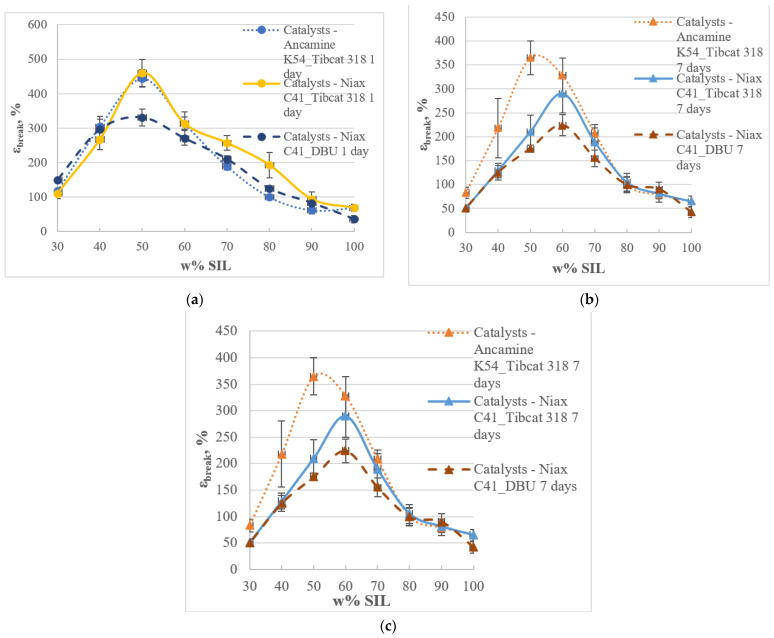
Tensile deformation (ε_break_) of the SIL/EP blends with different catalyst systems: Ancamine K54/Tibcat 318, Niax C41/Tibcat 318 and Niax C41/DBU after 1 (**a**), 7 (**b**) and 28 (**c**) days of curing at 23 °C and 50% RH.

**Figure 8 polymers-15-02269-f008:**
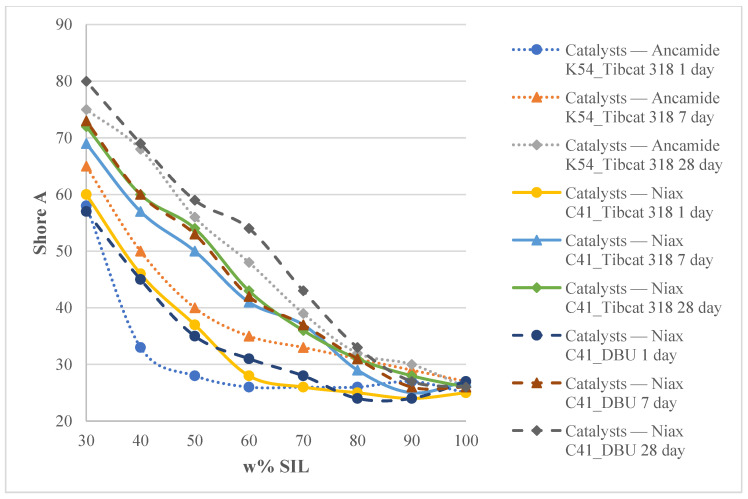
Hardness of SIL/EP blends with different catalysts systems: Ancamine K54/Tibcat 318, Niax C41/Tibcat 318 and Niax C41/DBU after 1, 7 and 28 days of curing at 23 °C and 50% RH.

**Figure 9 polymers-15-02269-f009:**
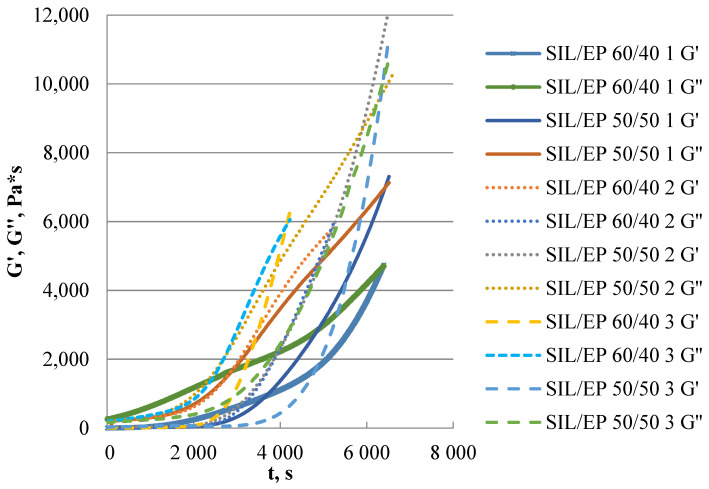
Elastic and viscous moduli growth in time. Catalyst systems are numbered as follows: 1—Tibcat 318/Niax C41 (solid line); 2—DBU/Niax C41 (dotted line); 3—Tibcat 318/Ancamine K54 (dashed line).

**Table 1 polymers-15-02269-t001:** Components of the two-component SIL/EP formulations.

Raw Material Class	Commercial Name	Raw Material Structure Formula
Prepolymer	SAX 520 (Kaneka, Belgium, Westerlo, Belgium)	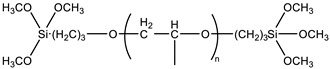
Prepolymer	D.E.R. 331 (Palmer Holland, Westlake, OH, USA)	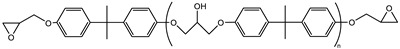
Compatibilizer	Dynasylan 1189 (Evonik Industries, Essen, Germany)	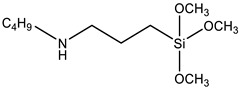
Catalyst	Tibcat 216 (TIB chemicals AG, Mannheim, Germany)	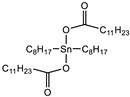
Catalyst	Tibcat 318 (TIB chemicals AG, Mannheim, Germany)	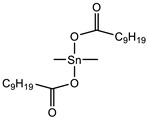
Catalyst	Tibcat 410 (TIB chemicals AG, Mannheim, Germany)	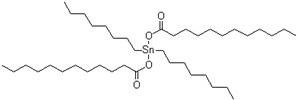
Catalyst	Niax C41 (Air products, Allentown, PA, USA)	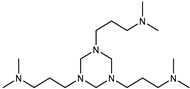
Catalyst	Dabco 33LV (Air products, Allentown, PA, USA)	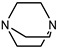
Catalyst	DBU (Air products, Allentown, PA, USA)	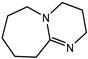
Catalyst	Ancamine K54 (Evonik Industries, Essen, Germany)	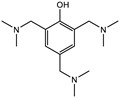
Catalyst	Jeffcat DMDEE (Huntsman Corporation, Conroe, TX, USA)	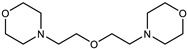
Catalyst	Tyzor TOT (The Shepherd Chemical Company, Cincinnati, OH, USA)	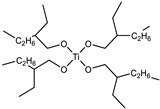
Catalyst	Bicat 8108M (The Shepherd Chemical Company, Cincinnati, OH, USA)	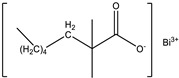
Catalyst	Bicat 3184 (The Shepherd Chemical Company, Cincinnati, OH, USA)	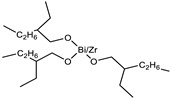

**Table 2 polymers-15-02269-t002:** Composition of two-component SIL/EP model system used for tensile and rheological tests.

	Raw Material Mass (g)							
SAX 520/D.E.R. 331	100/0	90/10	80/20	70/30	60/40	50/50	40/60	30/70
Dynasylan 1189	1.5	1.5	1.5	1.5	1.5	1.5	1.5	1.5
Tibcat 318	0.2	0.18	0.16	0.14	0.12	0.1	0.08	0.06
DBU	4	3.6	3.2	2.8	2.4	2	1.6	1.2
Ancamine K54	0	0.4	0.8	1.2	1.6	2	2.4	2.8
Niax C41	0	0.4	0.8	1.2	1.6	2	2.4	2.8
Water	0.2	0.18	0.16	0.14	0.12	0.1	0.08	0.06

**Table 3 polymers-15-02269-t003:** Elastic and viscous moduli values and crossover point data.

SIL/EP	1–60/40	1–50/50	2–60/40	2–50/50	3–60/40	3–50/50
Gc, (Pa)	4705	6425	5642	8647	5853	9931
tc, (s)	6395	6982	5168	5852	4142	6347

## Data Availability

The data presented in this study are available on request from the corresponding author.

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
