# Peer review of "Research of Potential Catalysts for Two-Component Silyl-Terminated Prepolymer/Epoxy Resin Adhesives"

_polymers, 2023, doi:10.3390/polym15102269_

Round 1

Reviewer 1 Report

The combining epoxy pre-polymer (EP) with a silyl-modified polyether (SIL) has a potential possibility to obtain materials with excellent toughness at a high strength and flexibility.Therefore, the research on this aspect has received wide attention. In this manuscript, the effects of different catalyst on the mechanical and rheological performance of the productions from individual and dual component of SIL and EP were investigated. Some conclusions obtained from this manuscript have  reference values to some extent for the study of adhesive materials. However, authors should consider the following suggestions.

1. Some problems related to reference. As an academic paper, some irformal literatures should be avoided to citated. For example, the reference [1]-[3],[11]. Additionally, some cited literatures do not correspond to the theme elaborated in the text. For example, the literature 14 is a platinum-containing catalyst rather than a tin-containing catalyst. Literature 18 has no relationship to SIL/EP. Line 41, “ In the current research attention is devoted to research...... [18], how does the 1983 literature stated the recent events. All the literature cited in the full text is in the introduction. This citation method is extremely hasty.

2. In the results and discussion, the obtained experimental results should be explained from the structural characteristics of the different catalysts used, rather than just the description of the phenomenon. For example, the different results caused by tin-containing catalysts or nitrogen-containing catalysts with different structure should be explained. Otherwise, the manuscript is an experimental report rather than an academic paper.

3. Line 101, DBU was tested at two catalyst concentrations of 0.5 and 4%... , However, in Table 2, there is not a concentration in 0.5%.

4. Line 96, “whereas the most effective was Tibcat 318 (dioctyl tin dicarboxylate).. , Does this sentence meet the result showed in Figure 1a.

5. In Figure 1b, whether the the curve for DBU 0.5% and curve for DBU 4% should be transposed.

Author Response

Dear Reviewer

Answers to your submitted questions are attached

Reviewer 2 Report

This manuscript studies the application of several metal Lewis acids and organic amines in the curing reaction of silane-terminated polymers with epoxy resins. Moreover, the rheological and mechanical properties of the cured crosslinked polymers were studied. This formulation system can be used for adhesives.  In this work, a systematic experimental study was carried out. However, this manuscript lacks innovation, because both metal Lewis acids and organic amine compounds are common catalysts for the curing reaction of epoxy system, which have already been written in many textbooks. The obtained adhesive has ordinary material properties. Therefore, I cannot recommend the publication of this manuscript.

Author Response

Dear Reviewer

Answers to your submitted questions are attached

Best regards

Reviewer 3 Report

This research focused on the investigation of potential catalysts for two-component Silyl-terminated prepolymer/epoxy resin systems. Characterization of the mechanical and rheological properties of epoxy adhesives has been conducted. In general, the findings indicate that it is possible to use traditional catalysts in individual systems, but certain alternative catalyst systems were found to be less toxic. Catalyst systems not only cure two-component systems, but also exhibit relatively high tensile strength and deformation values. This study opens the possibility for future research to reduce the chemical hazards of epoxy resin systems and to optimize their properties through catalyst selection. This paper reports a very nice compilation of experimental results and is worthy of publication in Polymers.

Below are a few points that I would like the authors to consider, in order to improve this manuscript.

1) The caption for Table 2 could be a bit more detailed. For example, please give a more detailed description of "m, g"; there is Dynasylan1189, but its structure is not listed in Table 1.

2) The description in lines 92-94 should be accompanied by a bibliography.

3) The description in lines 97-98 should be referenced.

4) The statement in line 102 should have a bibliography.

5) The mechanism (presumed mechanism) of the reaction that proceeds with the catalyst employed in this study should be shown in a chemical reaction equation.

In preparing such a diagram, please refer to Figure 1 of the following paper.

https://doi.org/10.1021/acsomega.1c05914

Author Response

(The authors gave the same response as above.)

Round 2

Reviewer 1 Report

The authors revised the manuscript following the recommendations proposed by reviews. Overall, revising the manuscript may be considered for publication. Please check the following suggestions.

1. Addtion of expoxy resin in keywords.

2. In line 89, water to catalyzeis not suitable. In fact, the role of water here is the reactant that initiates the hydrolysis.

3. The conclusions are too scattered, repeated and need to be further condensed. According to the research contents, the conclusions are suggested to be considered in three aspects from the catalytic point of view. (1)The rheological properties related to SIL and EP, (2)the mechanical properties related to SIL/EP two component system ; (3) the rheological properties related to SIL/EP two component system.

Author Response

Good afternoon

Comments were added in attached files.

Best regards
